# On the Origin of Plastic Deformation and Surface Evolution in Nano-Fretting: A Discrete Dislocation Plasticity Analysis

**DOI:** 10.3390/ma14216511

**Published:** 2021-10-29

**Authors:** Yilun Xu, Daniel S. Balint, Daniele Dini

**Affiliations:** 1Department of Mechanical Engineering, Imperial College London, London SW7 2AZ, UK; d.balint@imperial.ac.uk; 2Department of Materials, Imperial College London, London SW7 2AZ, UK

**Keywords:** discrete dislocation plasticity, contact, nano-fretting, size effect

## Abstract

Discrete dislocation plasticity (DDP) calculations were carried out to investigate a single-crystal response when subjected to nano-fretting loading conditions in its interaction with a rigid sinusoidal asperity. The effects of the contact size and preceding indentation on the surface stress and profile evolution due to nano-fretting were extensively investigated, with the aim to unravel the deformation mechanisms governing the response of materials subjected to nano-motion. The mechanistic drivers for the material’s permanent deformations and surface modifications were shown to be the dislocations’ collective motion and piling up underneath the contact. The analysis of surface and subsurface stresses and the profile evolution during sliding provides useful insight into damage and failure mechanisms of crystalline materials subject to nano-fretting; this can lead to improved strategies for the optimisation of material properties for better surface resistance under micro- and nano-scale contacts.

## 1. Introduction

Fretting refers to the process that causes relative tangential displacement of nano- and micro-meter amplitude between bodies in intimate contact [1] when cyclic loading is applied to the contacting surfaces. Fretting fatigue, which includes crack initiation and propagation, becomes a significant concern with materials designed for key components of some applications, such as titanium alloys applied in aerospace industry [2]. The resistance of a specimen surface to the imposed fretting fatigue is determined by the performance of the material and its resistance to shear. The presence of surface roughness and asperities in all real-life engineering applications causes the actual contact area to be only a small fraction of the nominal contact area [3]. Hence, it remains crucial to understand the contact size effect on the material’s behaviour under fretting fatigue when it comes to the length scale comparable to and below those characteristic of single asperities and the grain size polycrystalline materials [4,5,6,7].

Fretting fatigue crack initiation, which usually extends to a very significant proportion of the total fretting fatigue lifetime [5], has been extensively observed in aero and biomechanical applications [8,9]. Experimental work has shown the significance [10,11] of microstructure (e.g., grain morphology [12] and crystallographic orientation [13,14]) on the material performance under fretting fatigue. More specifically, the damage that eventually gives rise to the crack initiation has been associated with the highly localized plasticity at the length scale of single asperity contact size [15] and grain size [16], when the relative motion along contact is small. Therefore, the microstructure-sensitive modelling of fretting fatigue has been identified as a key vehicle to understanding the mechanistic driver [17] for crack initiation and lifetime prediction in fretting fatigue.

There has been a significant recent interest in the development of nano-fretting testing devices and experimental protocols [7,18,19] to investigate the evolution of the coefficient of friction (CoF) and surface roughness during fretting cycles. The average frictional force has been observed to reduce to a stable value with cyclic loading, and the surface roughness has been shown to either rise or drop depending on the ploughing effect of asperities. In spite of the conventional finite element modelling showing good fitting to experimental results in terms of fatigue life (e.g., [20]), the lack of a characteristic length scale of conventional plasticity prevents its use from accurately predicting the size effect in contact problems with applied tangential forces [21], especially when the asperity and contact size approaches the dislocation spacing [22]. Crystal plasticity finite element (CPFE) simulations [23,24,25,26,27,28] were employed to qualitatively reveal the correlation between damage and localized material deformation due to contact. However, the intrinsic continuum nature of the CPFE methodology and the necessity of fitting materials to constitutive laws’ parameters hampers its ability to address the mechanisms at the discrete dislocation scale [29]. The role of geometrically necessary dislocation (GND) has recently been reported as an important factor in multi-axial fretting [30] and fretting fatigue crack nucleation [13,14], which has motivated our investigation of the material’s response under fretting fatigue at the discrete dislocation scale. Previous analytical speculations [31,32] have established the potential link between micro-crack initiation of films under fretting to dislocation patterns and emissions near the contact point.

Deshpande and co-workers [21,33] performed monotonic micro-sliding discrete dislocation plasticity (DDP) simulations to investigate the dislocation structure and surface behaviour under sliding. Nicola and co-workers [34,35,36] have also employed the DDP framework to understand the activity of dislocations under multi-asperity, self-affine contact, and the roughness (i.e., the height of the asperities) has been shown to play a key role in the development of plasticity in the subsurface. To the authors’ knowledge, there is no computational framework published for fretting fatigue using DDP. Hence, no quantitative analysis has yet been reported on understanding the role of dislocations and the relation between surface damage [37] and contact size during fretting. A cyclic discrete dislocation plasticity framework is proposed herein for investigating the contact size effect on surface stress [38] and roughness [39,40], both of which are argued to be important factors in determining the lifetime of a specimen under fretting fatigue.

In this paper, a comprehensive set of DDP simulations were carried out to understand the contact size effect on the surface stress and profile evolution during nano-fretting. The simulation results show that shear stress and roughness development along the surface is governed by the contact size and, in turn, by how this affects dislocation activity during cyclic loading. The mechanistic understanding of the material resistance to nano-fretting can significantly aid the selection and optimization of surface design and engineering solutions.

## 2. Methodology 

### 2.1. Discrete Dislocation Plasticity Formulations

We used the planar, isotropic, isothermal discrete dislocation plasticity formulation that was firstly proposed by Van der Giessen and Needleman [41] in this study. The DDP formulations have been addressed in earlier articles, e.g., [33,42], and, hence, only key points are concisely summarised here. 

A single FCC crystal was applied to specimens, with the plane of simulation taken perpendicular to crystal direction [$\bar{1}0\bar{1}$] to satisfy the plane strain constraint. The material was assumed to be initially dislocation-free, and edge dislocations nucleated from Frank–Read sources, which were randomly populated along the slip planes with a predefined density in the specimen. Dislocation activities are governed by a set of constitutive laws, including mobility, pinning, and escape from obstacles, the details of which can be referred to [43]. Boundary conditions were satisfied using the superposition scheme [44], where the fields of displacement, stress, and strain were decomposed into a dislocation filed in an infinite elastic medium and a correction field. The former was obtained via summing up of analytical fields contributed to by all individual dislocations, while the latter was obtained via a finite element solution of a boundary value problem where singularities were absent and, thus, the dislocations’ effects were mediated by the corrected boundary conditions. 

In this research, there were two types of numerical simulations using discrete dislocation plasticity, namely: sinusoidal indentation calculations and nano-fretting calculations, the latter of which were realized by cyclically performing elementary nano-sliding calculations. Results obtained from two sets of sliding simulations were compared to reveal the effect of the preceding indentation load and corresponding contact size.

### 2.2. Sinusoidal Indentation Setup

Micro-indentation calculations were conducted on a film with thickness *H* = 10 μm (see Figure 1a) under a single sinusoidal-shaped asperity. The asperity shape was characterized with the wavelength λ = 10 μm and the amplitude Δ = 0.5 μm. Dislocation activity was confined to a process window of dimension *l* × *H* = 50 μm × 10 μm. The process window was bounded at both left and right sides to an elastic region. The total width of the film was chosen as sufficiently large at *L* = 1000 μm to avoid a boundary effect (i.e., trace surface condition at *x = ±L/2*). Inspired by [45], three slip systems with Φ^(α)^ = 0, ±45° with respect to the *x*-axis, respectively, were assigned within the DDP process window. Aluminium-like material properties were assigned to the specimen, whose parameters were identical those referred to in [33]. The dislocation source density that indicates the different initial status of materials (e.g., pre-strain [46,47], heat treatment, and pre-cracked, etc.) was fixed as ρ_nuc_ = 48.5 μm^−2^ to minimize the dislocation source starvation effect [48]. 

In sinusoidal indentation calculations, effects of geometry changes on the momentum balance and lattice rotations are neglected. However, the contact between the indented lower surface and film top surface is established on the deformed film surface. In an instant indentation process, the instantaneous applied indentation depth δ is imposed on the rigid-body indenter. The corresponding actual contact length *A* is defined as the range between the most left and right values of *x* coordinates where the indenter contacts the deformed top surface. In general, the actual contact length *A* differs from the nominal contact length $A_{N}=2\lambda \cos^{-1}\left(1-\delta /\Delta \right)$ due to sink-in or pile-up (see [49]), but it does not account for surface roughness (as analysed and discussed in [50]), which could lead to a significantly smaller contact area and, hence, spikes in indentation pressure due to random fluctuations along the contact, especially when sharp indenters are involved. The maximum indentation depth in this research was limited as δ_max_ = 0.2 μm, which was sufficiently small compared to the film thickness (relative indentation depth 0.02 only) to prevent the rigid substrate from taking its effect to disturb the film response [51] during the micro-indentation process.

The boundary conditions of the sinusoidal dentation problem are detailed as following:(1)$${\dot{u}}_{1}=0,{\dot{u}}_{2}=\dot{\delta }\;\mathrm{on}\;S_{contact}\;{\dot{u}}_{1}=0\;\mathrm{on}\;y=0\;\mathrm{and}\;{\dot{u}}_{2}=0\;\mathrm{on}\;x=0\;T_{1}=T_{2}=0\;\mathrm{on}\;y=H\;\notin S_{contact}$$
where *u_i_* is the displacement component, *S_contact_* the contacted fraction of the top surface, and $T_{i}=\sigma_{ij}n_{j}\;$the surface traction on a surface with normal vector *n_j_*. The displacement rate of the wedge-shaped indenter was set as$\;{\dot{u}}_{2}=\dot{\delta }=0.4\;{\mathrm{ms}}^{-1}$.

The reaction of film response to the indenter penetration is given by the surface traction:(2)$$F=-\int_{-A/2}^{A/2}T_{2}\left(x,H\right)dx$$

The actual indentation pressure *p_A_* is calculated by: (3)$$p_{A}=F/A$$

### 2.3. Sliding and Fretting Setup

The fretting simulations were composed of a number of elementary reciprocating sliding simulations, the setup of which is described below. The specimen dimension and the material properties used in the following sliding and fretting calculations are identical to those used for the indentation cases (see Figure 1b). The contact between the sinusoidal asperity and specimen was modelled via a resistant adhesion zone on the contacting surface of actual length A with a relation between shear traction versus displacement, which is given by: (4)$$T_{t}=\left\{\begin{array}{c} -\tau_{max}\frac{\Delta t}{\delta_{t}},if\;\left|\Delta t\right|<\delta_{t}\\ -\tau_{max}sign\left(\Delta t\right),if\;\left|\Delta t\right|>\delta_{t} \end{array}\right.$$
where $\Delta t=u_{1}\left(x,H\right)$ is the tangential displacement jump across the cohesive surface, δ_t_ the critical jump, τ_max_ is the cohesive strength, and *T_t_* represents the shear traction response. The contact condition that considers the pure shear without normal preceding indentation represents a light tribological loading [33]. The contact could be considered as penetrable or impenetrable to dislocation passing through, the latter of which was applied in a molecular dynamics study [52]. The study indeed showed the effect of both surfaces being “penetrable” and “impenetrable” to dislocations, and this would depend on the local constraints and specific material properties of the contacting bodies. Here, we considered the surface to be permeable to dislocations, in view of the fact that fretting experiments have shown that some evolution of surface roughness [19] underneath the contact is possible.

The maximum cohesive strength was set to τ_max_ = 300 MPa and the threshold displacement jump was δ_t_ = 0.5 nm, values we have used in our previous work [53] when we explicitly explored the origin of microstructural changes under various sliding and loading conditions. The non-softening form of the traction was adopted to help convergence of surface tractions. The temperature rise due to frictional work and the temperature influence on dislocation activity [47,54] was neglected in this research and a future separate study will be devoted to investigating this effect.

The sliding rates,
(5)$${\dot{U}}_{x}=\dot{U},\;{\dot{U}}_{y}=0$$
are imposed on the specimen boundaries *x* = ±*L*/2 and *y* = 0 to simulate the relative motion of contact surface with the rate $\dot{U}/A=10^{4}s^{-1}$, which substantially rules out the sliding rate sensitivity [55]. 

The averaged shear stress τ along the contact is given by:(6)$$\tau =-\frac{1}{A}\int_{-A/2}^{A/2}T_{x}\left(x,H\right)dx$$

In one set of sliding calculations, the film was subjected to a pure shear slide condition i.e., with an initial status as an indentation-free (and hence dislocation- and stress-free) condition. Shear stress, along with predefined contact, was studied without normal stress in [21,33,55]. However, this set of calculations can only approximate the sliding scenario as the local material deformation near the contact surface and subsurface, by virtue of the application of the normal load, has not yet been taken into consideration. Issues, including surface elevation [33] during sliding produced by the absence of normal stress, also limit the application of pure shear sliding. In the second set of sliding calculations, a sinusoidal indentation simulation was firstly carried out to establish contact, the initial dislocation structure, and initial stress and strain fields produced by a certain indentation depth before performing calculations for the forthcoming fretting models. The latter set of simulations aimed to capture more realistic features of material deformation under fretting in the presence of an applied normal load and initial dislocation structures, the effect of which has been comprehensively investigated and shown to be very important in the context of indentation pressure response during micro-indentation [51]. 

Nano-fretting was realised by performing the reciprocating sliding calculations [56] with a sliding stroke, *U*, ranging between 0 and 20 nm. Results were sampled when the sliding reached the maximum stroke and returned to the initial point (corresponding to a tangential load ratio *R = 0*) to understand the contact size effect in determining the shear stress and the surface profile variation during fretting.

## 3. Surface Stress and Profile Evolution during Fretting 

The evolution of the calculated shear stress, τ, averaged over the contact length is shown as a function of the applied sliding distance *U* in Figure 2 for the contact size *A* = 2.0 μm under fretting without a preceding indentation. For all cycles, the shear stress curve during the loading (sliding forward) half-cycle was systematically higher than the one during corresponding unloading (sliding backward) half-cycle. This behaviour is similar to the well-known Bauschinger effect of crystalline materials under a cyclic uniaxial loading [57]. In addition, the average shear stress exceeded the nucleation strength of dislocation sources when contact approached the end of the fretting stroke. This phenomenon indicates that the film had been plastically deformed, even during the first forward cycle, which corresponds to dislocations piling up and other associated activities. The area enclosed by a pair of loading and unloading curves represents the overall energy dissipated during a fretting cycle. The dissipated energy was found to become smaller with fretting cycles, which indicates that plastic shakedown [58] has generally occurred.

The surface profile (corresponding to the y-component of nodal displacements along the contact surface) variation during the fretting simulation is shown in Figure 3a for the contact size *A* = 2.0 μm. Results were obtained at the end of each indicated cycle, i.e., when the contact returned to the initial point. The surface profile generally showed a tendency to “bulge out” during the fretting process due to permanent deformations due to the activity of dislocations that caused microstructural and nano-/microscale roughness rearrangements at the surface [59], as well as the fact that normal pressure was absent during simulations; such loading scenarios are probably more typical of contacts undergoing sliding in the presence of adhesive forces rather than intimate contact induced by the application of normal load [19]. These results will be further compared to the results obtained with preceding indentation. The profile variation was generally distributed uniformly across the contact surface. However, at the edge of the contact there tended to be abrupt discontinuities that were produced by material intrusions and/or extrusions. These were argued to originate from certain dislocation pile-up patterns and dislocations exiting along favourable slip systems [60], such that dislocation steps were left along the contact surface. The detailed dislocation structure that developed near the contact edges at the end of 50th cycle is presented in Figure 3b for the contact size *A* = 4 μm. The subset of dislocations that exited from the contact surface are labelled using a dashed symbol along the inclined slip systems. Dislocation ejections, re-arrangements, and pile-ups were found to be the mechanistic drivers for the surface profile discontinuity localized near the contact edges.

## 4. Contact Size Effect 

Fretting simulations with the maximum stroke distance of 20 nm are carried out with contact size of *A* = 2, 4 and 10 μm, respectively, without preceding indentation. The peak average shear stress along the contact, which was calculated when the contact reached the maximum sliding distance, is reported in Figure 4a for the three contact sizes. At earlier cycles, the shear stress exhibited a contact size dependence that has been revealed in previous numerical studies [21] and experimental evidence [61]. The shear stress was dominated by the cohesive strength for a small contact size, whereas it was governed by the dislocation source strength (*τ**_0_* = 50 MPa) for large contact sizes. However, this dependence gradually diminished with the cycle number in the long-term and, as a result, the shear stress discrepancy between the two extreme contact sizes reduced from 60 MPa to 15 MPa after 100 cycles. For a fixed maximum stroke length, the small contact size appeared to be more sensitive to the application of the cyclic loading when plasticity was introduced into the superficial layer during fretting. For these cases, in the scenario characterised by the small area of contact, the plastic flow extended more into the region adjacent to the contact (while for larger contacts, the new area of material coming into contact was relatively small compared to the contact area). It is shown that the stress decreased and would eventually achieve a contact-insensitive value, which can be argued to be linked to the dislocation source strength as a measure of the material resistance to flow at the macroscale. The evolution of shear stress is in line with other analyses performed at the continuum length scale [62], which showed that surface stress reduces with fretting cycle number. This is due to the inhomogeneity of the stress field under the contact and the fact that shear stresses are highly localized at the contact edges during the initial cycles and gradually become more homogenously distributed along the contact. The arithmetic average, *R_a_*, of roughness along the contact that was induced by the fretting loading is shown in Figure 4b. The surface roughness that was produced by dislocations piling up and exiting from the surface [31] developed further with an increasing number of cycles [19]; this can be argued to be one of the important factors determining fatigue crack initiation at the contact edge, as observed in the majority of fretting problems, especially those characterised by sharp-ended contacts [63,64,65]. The rate of change of roughness was inversely proportional to the contact size. For a given maximum stroke distance, a small contact size results in a stress concentration [66] and sharp stress gradients near the contact. Hence, dislocations are more likely to leave the system from the contact surface, where the majority of the plastic deformation occurs. The displacement steps left by them at the surface accumulate and eventually form the roughness as perceived at the macroscopic scale. The influence of the contact size on the evolution of the surface roughness can also be linked to the effect that realistic roughness, where multi-asperity contacts must be considered, has on fretting life and crack initiation [67]. The combination of the dependence of shear stress and surface roughness on contact size can also provide a rigorous mechanistic explanation for the fatigue life dependence on contact size [66,68,69] when contact and slip are confined within one asperity length scale.

## 5. Preceding Indentation Effect

The results that are reported above were based on calculations without prior indentation and application of normal contact load; this is shown to lead to surface elevation and specific patters of roughness evolution during fretting (please see Figure 3a) in the absence of indention pressure. The following paragraphs discuss the surface profile and shear stress evolution, respectively, during a fretting process with a preceding indentation. Calculations aimed to reveal the indentation effect that is commonly associated to practical fretting applications (e.g., [4,70]). The number of cycles was limited to three due to computational expense and the fact that the system seems to shakedown to a stable solution within the first few cycles. Shear stress τ averaged over the contact evolution that was obtained from fretting calculations with a preceding indentation process as a function of the sliding distance, *U*, is plotted in Figure 5. Results are shown for three contact sizes: (a) *A* = 2.4 μm, (b) *A* = 4.1 μm, and (c) *A* = 9.3 μm, which were established by the preceding indentation process. It can be seen that the abovementioned Bauschinger effect diminished in fretting with preceding indentation compared to the results in the absence of indentation pressure. In addition, the overall energy dissipated during each fretting cycle decreased with increasing contact size. This phenomenon is also attributed to the presence of the normal indentation pressure, which, by producing extensive dislocation activity (see [71]), dominates the process, introducing plastic flow and permanent deformations just underneath the contact. Fretting loading with such a small sliding stroke length was shown not to be able to contribute to cyclic damage in the presence of relatively large contacts.

The surface profile change was again investigated under three similar contact sizes *A* = 2.4, 4.1, and 9.3 μm, which were established by the preceding indentation. The surface profile at the instant the slider returned to the initial position under contact size *A* = 2.4 μm is shown in Figure 6a for the first four cycles. The surface profile (represented by the *y*-displacement) remained similar, in terms of general shape, to the profile established by the preceding indentation (note the *x–y* axes ratio is not in scale); the application of nano-fretting cycles relieved the stress field, such that the surface profile appeared to recover slightly towards the undeformed configuration, for both of the sink-in and pile-up partitions identified in the inset to Figure 6a. This recovery was also observed in the other two contact sizes and manifested itself in the variation (slight reduction) of the surface roughness, represented by arithmetic average *R_a_*, in the subsequent nano-fretting cycles, as reported in Figure 6b. Hence, for the conditions studied here, the surface roughness during nano-fretting was governed by the preceding indentation, which resembles the findings from the fundamental wear rate model [72]. A large contact size that is introduced by a large proceeding indentation has been shown to result in large surface roughness, which would in turn lead to earlier fretting fatigue crack nucleation [15].

In order to further investigate the difference between pure sliding and sliding with preceding indentation, the dislocation structure and corresponding slip contours are plotted in Figure 7. Results are shown for sliding for a h = 10 μm film for U = 0.02 μm and a≈10 μm; the preceding indentation was performed with the sinusoidal asperity. The dislocation structures were quite different between the two simulations. Dislocation structures obtained from a pure sliding calculation typically showed a “strip” of dislocation activity (whose width is dependent on the contact area) just underneath the contact surface of the film. The dislocation patterns induced by sliding with a preceding indentation were affected by the structure generated by initial normal load and persisting after indentation, which led to more extensive and dense structures away from the contact.

## 6. Conclusions

Nano-fretting analyses were carried out on the discrete dislocation scale to extensively investigate the shear stress and surface profile evolution of a single crystal contacted by a single asperity. Several conclusions can be drawn as follows: The average shear stress along the surface was reduced through dislocations gliding, while surface roughness was introduced and produced discontinuities at the contact interface. These phenomena both evolved during fretting cycles and tended to stabilise after a few cycles.For contact problems without preceding indentation, the evolution of the shear stress and roughness was shown to be more sensitive to contact size.For scenarios in which contact was established with preceding indentation, a large contact size was shown to lead to more severe dislocation accumulation at the contact edges, which in turn could be one of the possible precursors for driving fatigue crack initiation during nano-fretting. However, this needs to be considered together with other relaxation mechanisms, which could counteract the tendency to nucleate cracks at regions of large surface curvature generated by large GND density.

The methodology presented in this article can also be used to study phenomena linked to near-surface microstructural evolution in the context of nano-indentation, such as [73,74]. In addition, the research sheds lights on the surface evolution during tribological loading [75] and reveals the significant contribution of dislocation activities near the contact surface [76,77] to the microstructure evolution.

## Figures and Tables

**Figure 1 materials-14-06511-f001:**
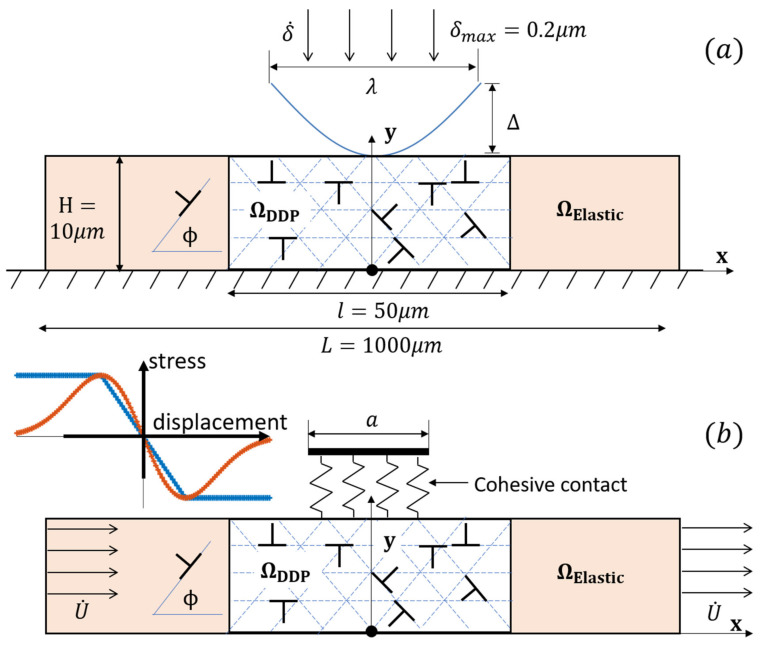
Schematic diagrams of the (**a**) sinusoidal indentation and (**b**) nano-sliding and fretting boundary value problems analysed using a discrete dislocation plasticity framework. The origin of the coordinate system employed is marked as a filled-in circle (•) in (**a**) and (**b**). The softening and non-softening cohesive relation between shear traction and jump displacement on contact partition is illustrated in (**b**). The non-softening relation is utilized for the fretting calculations.

**Figure 2 materials-14-06511-f002:**
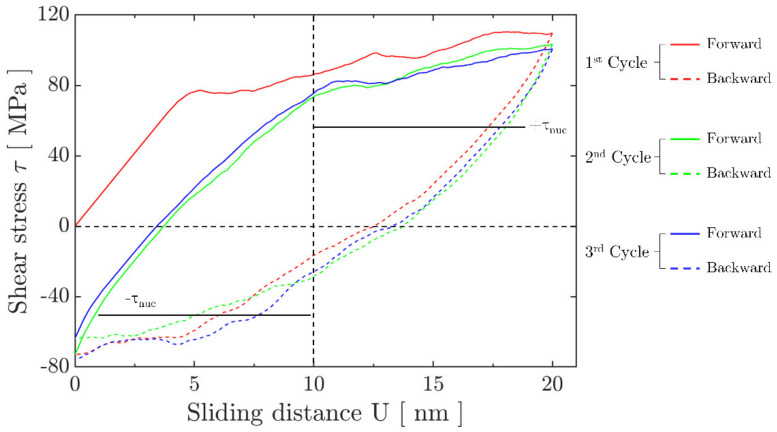
Average shear stress τ evolution during early cycles of a fretting process. Results were obtained by cyclic sliding for a h = 10 μm film with a stress- and dislocation-free status initially. The contact length is set as *A* = 2.0 μm.

**Figure 3 materials-14-06511-f003:**
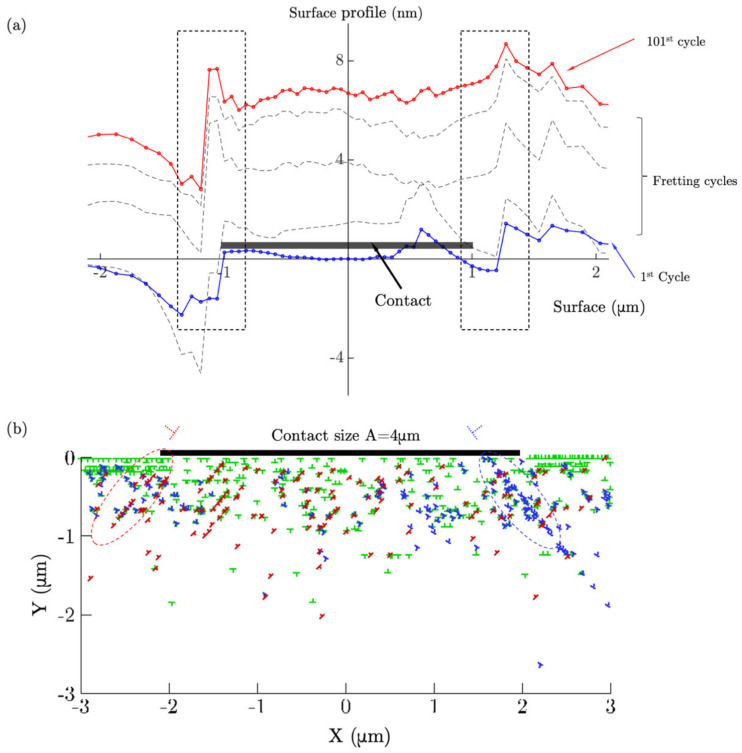
(**a**) The surface profile along the contact surface (the contact length was set as *A* = 2.0 μm) after 100 cycles without a preceding indentation. (**b**) The instantaneous dislocation structure after 50 cycles for contact size *A* = 4.0 μm.

**Figure 4 materials-14-06511-f004:**
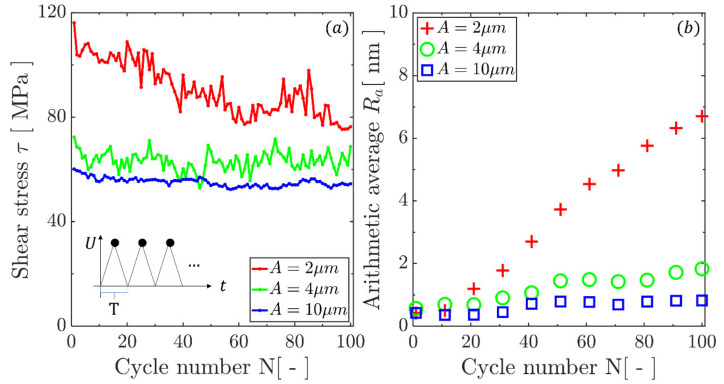
Cyclic behaviour of (**a**) peak shear stress that is obtained at the maximum sliding distance of a loading cycle and (**b**) roughness along the contact surface (represented by arithmetic average *R_a_*) under three different contact sizes. Results were obtained from fretting calculations without preceding indentation.

**Figure 5 materials-14-06511-f005:**
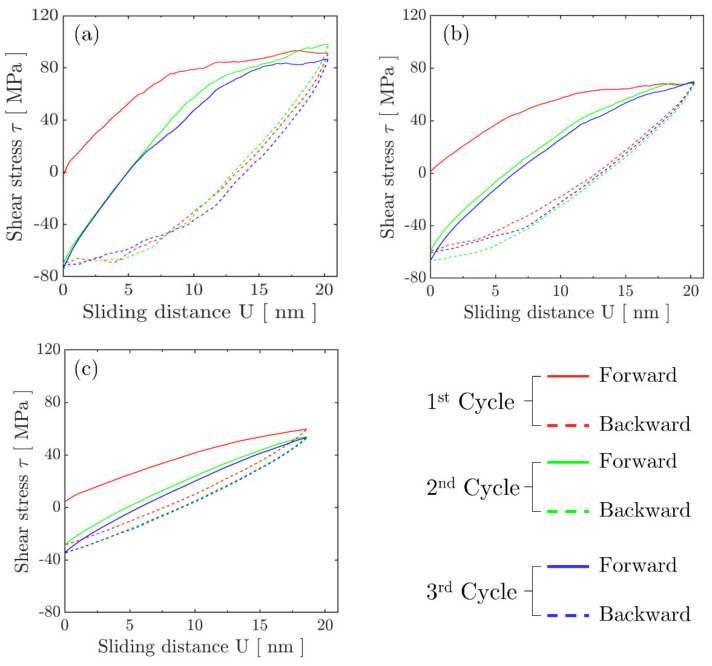
The shear stress evolution along the contact surface during the first 3 cycles for contact size (**a**) *A* = 2.4 μm, (**b**) *A* = 4.1 μm, and (**c**) *A* = 9.3 μm.

**Figure 6 materials-14-06511-f006:**
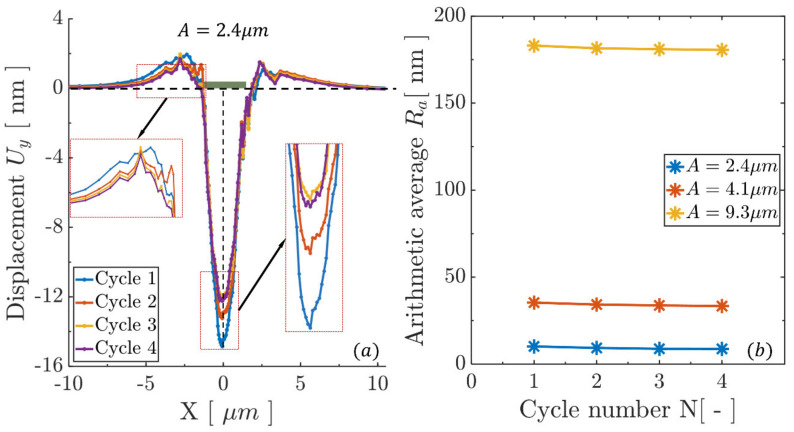
Preceding indentation effect on surface roughness evolution with cycles: (**a**) the surface profile under contact size *A* = 2.4 μm, which was introduced by the preceding indentation with 4 cycles, (**b**) roughness evolution with cycles under the three contact sizes.

**Figure 7 materials-14-06511-f007:**
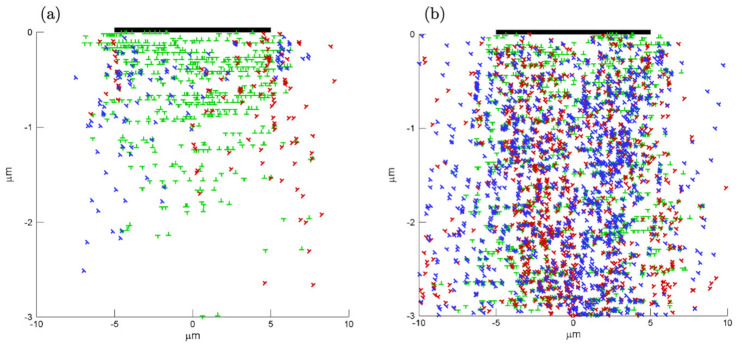
Effect of preceding indentation on dislocation structure. Dislocation patterns are obtained for U = 0.02 μm of simulations (**a**) pure sliding with contact size a = 10 μm (**b**) sliding with a preceding indentation that introduces a contact size a ≈ 10 μm.

## Data Availability

The data presented in this study are available on request from the corresponding author.

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
