# Peer review of "On the Origin of Plastic Deformation and Surface Evolution in Nano-Fretting: A Discrete Dislocation Plasticity Analysis"

_materials, 2021, doi:10.3390/ma14216511_

Round 1

Reviewer 1 Report

The paper concerns the origin of damage and surface evolution in nano-fretting. The research topic is very interesting, the authors showed their contribution to the development of modelling of nano-fertting. In the introduction, the authors analyzed in details the literature on research topics. On this basis, they clearly formulated the aim of the work, which shown their contribution to the development of this research area. The presented research methodology is clear and describes the simulations carried out in detail. The figures presented in the work clearly show the obtained results. The conclusions summarize the most important research results presented in the manuscript

To conclude, I recommend this paper for publication in a present form.

Reviewer 2 Report

The authors carried out 2D DDD to study the nano-fretting process of a single crystal. The study is built upon the previous work of Deshpande et. al where the effect of indentation on the following sliding was studied.

I have several questions/comments for the authors to consider:

1, As the author mentioned in Figure 3(a), it does not really sound right to have just pure shear without normal, simply because the fact that the surface becomes rough at the contact would destroy the contact and adhesion. There was MD study in Lucia Nicola's group that shows the contact interface is 'impenetrable' for dislocations, so the authors could just allocate impenetrable obstacles underneath the contact, then look at the surface profile evolution outside the contact.

2, the explanation of the contact size effect (figure 4) does not really reach the point. The author should show the plot of dislocation density defined in a window that is related to contact size to see how it evolves with the increasing cycles. I believe in the dislocation density regime the author studies, the increase of the dislocation density would result in a decrease of the yield stress.

3, The authors should discuss the results of indentation, otherwise how it is possible to see the connection with the following fretting analysis. Show the dislocation density in a defined window. In such a study, dislocation density is important, unfortunately, the authors did not mention it at all.

4, regarding conclusions, unfortunately, I have questions/comments on each of them. 

4.1 The average shear stress along the surface is reduced while surface roughness is introduced through dislocations piling up.  This is wrong. The stress is reduced because of the dislocation activity, and in fact, dislocation gliding. Dislocation pileup would increase the backstress and plays the role of hardening....

4.2 For contact problems without proceeding indentation, a smaller contact size is shown to be more sensitive to changes in shear stress and roughness evolution. This sentence has a logical problem.....the smaller contact size is the smaller contact size, does not depend on anything....it is the evolution of the shear stress and roughness being sensitive to the contact size. 

4.3 For scenarios in which contact is established with proceeding indentation, a large contact size is shown to lead to more severe dislocations accumulation at the contact edges and increased roughness, which in turn could be responsible for driving fatigue crack initiation during nano-fretting. This seems to contradicts with what figure 6 says. First of all, it was said that roughness is mainly determined by the indentation profile. Second, the authors did not show severe dislocation accumulations at all.

Reviewer 3 Report

Review of the article

 «On the origin of damage and surface evolution in nano-fretting: a discrete dislocation plasticity analysis»

Investigation of the processes occurring at the boundary of two bodies when sliding relative to each other is an urgent and difficult task. In the initial stages, plastic deformation processes take place. The article is devoted to the numerical simulation of the interaction of a rigid (non-deformable)  sinusoidal asperity  with the crystal surface at this initial stage. The crystal is plastically deformed by dislocation slip. The model of discrete dislocation slip proposed by other authors is used. In the work under review, the model was generalized for multiple interactions. Plastic deformation of crystals, at least in the initial stages, is determined by the motion of individual dislocations. Therefore, the model used in the work is scientifically sound. The evolution of the surface with an increase in the number of cycles is studied. Previously, such models were not used in the analysis of cyclic deformation modes, which determines the novelty of the work. In this part, the results are original. But, as with every numerical simulation work, the results are highly dependent on the process model.

The work was performed at the level we currently exist in the field of numerical modeling of plastic deformation processes. It is not outstanding, but it is not frankly weak either.

The text of the article is clear and easy to read.

The conclusions are consistent with the results obtained. But the phrase "... which in turn could be responsible for driving fatigue crack initiation during nano-fretting" has not been proven and does not follow from the results of the work. Surface curvature may not be accompanied by crack formation (in the presence of other relaxation mechanisms).

The title of the article does not reflect the content of the work. The nature of the origin of damage is not considered in the work.

Round 2

Reviewer 2 Report

I appreciate the changes the authors made, I have no further comments or suggestions.